# Effects of an Extracurricular Anger Self-Control Program for Nursing Students

**DOI:** 10.3390/ijerph18063059

**Published:** 2021-03-16

**Authors:** Won Hee Jun, Eun Joung Choi, Hyun-Mee Cho

**Affiliations:** 1College of Nursing, Keimyung University, 1095 Dalgubeol-daero, Dalseo-gu, Daegu 42601, Korea; jwh9178@hanmail.net; 2College of Nursing, Kosin University, 262 Gamcheon-ro, Seo-gu, Busan 49267, Korea; 3Department of Nursing, Kaya University, 208 Samgye-ro, Gimhae, Gyungnam 50830, Korea; yuchanmom@nate.com

**Keywords:** anger, anger expression, nursing student, grateful disposition, depression

## Abstract

Nursing students often experience anger in response to stress and suppress their anger instead of actively controlling it. Therefore, the anger self-control programs that can manage nursing students’ anger level and dysfunctional anger expression are needed. The aim of this study is to investigate the effects of an anger self-control program on trait anger, anger expression style, grateful disposition, and depression among nursing students. The study used a quasi-experimental study with a nonequivalent control group and a non-synchronized design. Participants were 29 nursing students who were assigned to intervention and control groups. Compared to the control group, the intervention group showed significantly decreased mean scores for the trait anger, anger-in, and anger-out anger expression styles, and increased mean scores for the anger-control anger expression style and grateful disposition. Anger self-control programs might be usefully applied as extracurricular anger-management programs for nursing students.

## 1. Introduction

Violent behavior- and anger-management-related problems in school have rapidly increased among young adults worldwide [1,2]. In Korean society, where a collectivist culture is deeply embedded, expressing one’s emotions or thoughts is considered to represent excessively self-assertive behavior and, thus, it is negatively viewed; consequently, individuals tend to suppress their anger [3]. Anger or dysfunctional anger expression in college undergraduates are important issues that require active attention, as they are intimately related to serious mental health problems, such as suicidal ideation [4], depression [5], interpersonal relations problem [6], and addiction [7], as well as to antisocial behaviors, such as dating violence [8].

Anger is especially a common emotional problem experienced in response to stress among nursing students [2]. A previous study found that nursing students have a higher level of anger than students of other majors, and that they often suppress their anger and address it independently, as opposed to actively controlling and resolving it [9]. Anger and inappropriate anger control in nursing students may negatively influence the quality of care that they provide [2,10]. In this regard, anger management skills are essential for nursing students, as occasional conflict is common in clinical situations. Thus, nursing education should be modified to more actively help students accurately recognize their anger and the issues associated with their anger expression methods, as well as to improve their ability to control anger, which would help them express their anger appropriately.

A previous study that analyzed the effects of anger-management programs conducted in Korea over the preceding 12 years reported that the effects of anger-management programs are most frequently analyzed among adolescents, and least frequently among college students [11]. Further, the study stated that past studies only focused on verifying the effects of the programs, without clearly considering the programs’ accessibility and sustainability [12]. Considering this, it is clearly important to develop anger self-control programs (ASCP) for nursing students and to devise measures for improving students’ access to these programs and for increasing the sustainability of the programs.

Nursing education consists of curricular and extracurricular components. Extracurricular programs, such as coaching, mentoring, counseling, and club activities, supplement college curricula by ensuring diversity and promoting voluntary student participation [13]. Furthermore, professor–student interaction is an important educational experience that improves students’ self-control behaviors, intellectual growth, and character [13]. Therefore, it is imperative that colleges offer programs in which students and professors can interact outside of lecture settings, as this would help students adapt to college life. In a study in which focus group interviews were conducted with Korean college students, Jun, Jo, and Jeong [14] suggested that professors with expertise in anger-management and who worked in a department of nursing should be allocated to anger-management programs for nursing students, as this would increase students’ access to the programs and contribute to ensuring that the programs run sustainably. Therefore, establishing ASCPs as extracurricular programs managed by the professors may be an effective strategy for running these programs.

Most previous anger-management programs for college students have been based on the cognitive behavioral model, and recent studies have incorporated arts, psychological approaches, and mindfulness meditation into such programs [1,15,16,17]. Anger is a concept that encompasses interactions between cognitive, emotional, and behavioral components, and specifically addressing these components, as well as strengthening effective coping behaviors for anger overall, may be important. Recent advances in positive psychology have highlighted that positive psychological features can constitute protective factors against anger [18]. Considering this, the present study developed an ASCP that differs from existing programs by focusing on increasing understanding of the emotional, cognitive, and behavioral aspects of anger, based on the cognitive behavioral model, and on strengthening coping behaviors for controlling anger effectively, based on the positive psychological model as an extracurricular program. Moreover, the effects of developed ASCP were assessed by grateful disposition reported to have protective effects against anger [19] and a risk factor, i.e., depression for anger expression [20], as well as trait anger and anger expression style. The aim of this study is to investigate the effects of an anger self-control program on trait anger, anger expression style, grateful disposition, and depression among nursing students.

## 2. Methods

### 2.1. Design

This study was a quasi-experimental study that used a nonequivalent control group and non-synchronized design. The independent variable was six sessions of an anger self-control program (ASCP). The dependent variables were trait anger, anger expression style, positive thinking, and grateful disposition.

### 2.2. Participants

Twenty-nine second-year nursing students were conveniently sampled from K university in P city, South Korea. The inclusion criteria were: (1) individuals who had never participated in an anger-management program, (2) individuals who had never undergone drug therapy or psychological counseling for a mental health problem, and (3) individuals with a mean score for trait anger exceeding the median (2.5) as measured by the Korean version of the State-Trait Anger Expression Inventory (STAXI).

Nursing departments that involve clinical training begin this training from the third year, so it is difficult for third- and fourth-year students to participate in extracurricular programs in school. For these reasons, second-year nursing students may be the most appropriate target of extracurricular ASCP.

Sample size was calculated using the G Power 3.1 software (The G*Power Team, Kiel, Germany). The minimum sample size for each group was determined to be 12: this was obtained using an effect size of 0.35 from previous studies [10], a significance of 0.05, a power (1-β) of 95%, and a repeated-measures analysis of variance (ANOVA). However, considering a 20% dropout rate, we recruited 30 participants in total, allocating 15 to each group. The participants were assigned by a draw method of randomization to the intervention and control groups based on their classes. This was in order to prevent intervention contamination. Specifically, of the two second-year classes, one (*n* = 15) was assigned to the intervention group and the other (*n* = 15) was assigned to the control group. One student in the intervention group took a leave of absence from school before the end of the program and, thus, was withdrawn from the study. As a result, 29 participants (14 in the intervention group and 15 in the control group) participated in this study.

### 2.3. Procedures

The ASCP was developed based on the cognitive behavioral model, positive psychological model, and previous studies that applied anger-management programs for adolescents and undergraduates [15,21,22]. The contents of this program were structured based on consultation with one psychiatrist, two psychiatric nursing professors, one psychology professor, and one education professor, and these professionals also validated the program; at this stage, the content and composition of each item was rated using a four-point scale (4 = very valid, 3 = valid, 2 = not valid, 1 = not valid at all), and the content validity index (CVI) was 0.89.

As an extracurricular program for nursing students, it would be most desirable to administer ASCP in a manner that causes little burden on students’ studies and professors’ work, as this would increase the accessibility and sustainability of the programs. A previous study [1] verified the effects of anger-management programs for college students when a minimum of three sessions are applied; however, this was conducted overseas, and a similar study conducted in Korea [12] stated that six sessions are necessary; consequently, the ASCP in this study was designed as a six-session program.

Improving college students’ self-control capacities and their abilities to promote psychological adjustment involves helping them to recognize negative emotions related to stress and distorted cognitive processing, and to strengthen their behavioral coping techniques [23]. From this perspective, this program was developed with a focus on the following two topics: (1) “understanding anger”, which concerns accurately understanding the fundamentals of anger in terms of various emotions and cognitive distortion, as well as appropriate expressions of anger, and (2) “placating my angry self”, which concerns useful behavioral strategies for effective anger control. “Understanding anger” was addressed in four sessions. The first session concerned emotion differentiation, which refers to specifically and clearly identifying emotions. The second session focused on recognizing the fundamentals of anger by exploring various emotions and needs related to anger. The third session enabled participants to learn that anger is an individual’s choice—this was achieved through reviewing cognitive distortion that induces anger and thoughts that may pacify anger. Finally, in the fourth session, participants explored the positive and negative outcomes of anger expression and learned about healthier ways of expressing anger. Meanwhile, “Placating my angry self” consisted of two sessions (sessions 5 and 6). In the fifth session, participants performed activities to strengthen positivity. Then, in the sixth session, participants were trained to control their minds by expressing empathy for the emotions and needs of others involved in anger situations and by forgiving others.

Each session consisted of a warm-up (20–30 min), activity (40–50 min), and follow-up (20 min). In the warm-up, changes since the last session and homework from the previous session were reviewed. In the activity section, participants performed activities related to the topic and received instruction in small groups of four to five or individually. In the follow-up section, participants shared their thoughts on the activity and were given feedback on their performance and an explanation of the week’s homework. The details of the program are shown in Table 1.

The ASCP was conducted once a week, with each session lasting 90 min. In order to complete the program and a second post-test held during the semester, the intervention group performed the program from weeks 2–7 in the spring semester, at the seminar room of college where the principal investigator works, every Wednesday. The program was administered by the principal investigator, who was a licensed (class 1) mental health professional and psychiatric nurse specialist who had taught students in college for 10 years and had administered problem-solving-focused group education and counseling programs for alcoholic patients in the psychiatric ward of a university hospital for ten years. In addition, this investigator had completed a basic and intermediate cognitive behavioral therapy program, an anger management workshop, an empathetic conversation workshop, and a positive psychology strength professional certification program.

### 2.4. Measures

#### 2.4.1. Trait Anger and Anger Expression Style

Anger was measured using the Korean version of the State-Trait Anger Expression Inventory (STAXI), which was originally developed by Spielberger, Krasner, and Solomon [24] and was translated into Korean and modified, verifying its validity and reliability, by Chon, Hahn, Lee, and Spielberger [25]. This inventory consists of 20 items for measuring state and trait anger (10 each), which concern the empirical aspect of anger, and 24 items for measuring anger expression styles (anger-in, anger-out, and anger control—eight items each). In the present study, we used 10 items for measuring trait anger, which measured individuals’ general anger tendencies, and 24 items for the anger-in, anger-out, and anger control scales. Each item was rated using a four-point Likert scale (1 = not at all, 4 = very much so), and a higher trait anger score indicated a higher level of anger. Further, a higher score for each of the anger expression scales indicated a higher degree of use of the corresponding anger expression style. The reliability (Cronbach’s α) of the tool in Chon et al.’s [25] study was 0.90 for state anger, 0.82 for trait anger, 0.73 for anger-in, 0.74 for anger-out, and 0.81 for anger control. Meanwhile, the reliability (Cronbach’s α) of the tool in our study was 0.87 for trait anger, 0.76 for anger-in, 0.78 for anger-out, and 0.80 for anger control.

#### 2.4.2. Grateful Disposition

Grateful disposition was measured using the Korean version of the Gratitude Questionnaire-6 (GQ-6), which was originally developed by McCullough, Emmons, and Tsang [26] and was translated into Korean and adapted and validated for use on college students by Kwon, Kim, and Lee [27]. This six-item tool is rated using a seven-point Likert scale. A higher score indicates a more grateful disposition. The reliability (Cronbach’s α) was 0.85 in Kwon et al.’ [27] study and 0.90 in this study.

#### 2.4.3. Depression

Depression was measured using the Korean version of the Beck Depression Inventory-II (BDI-II). The original BDI-II was developed by Beck, Steer, and Brown [28] and later translated and adapted into Korean and validated by Sung et al. [29]. This inventory consists of 21 items, and each item is rated using a four-point Likert scale (0–3), where higher scores indicate higher levels of depression. The score breakdown is as follows: a total score of 0–13 equates to “no depression at all or very mild depression”, 14–19 equates to “mild depression”, 20–28 equates to “moderate depression”, and 29–63 equates to “severe depression”. The reliability (Cronbach’s α) was 0.83 in Sung et al.’s [29] study and 0.90 in this study.

### 2.5. Data Collection

Data were collected in the spring semester of 2019 after obtaining approval from the Institutional Review Board (IRB) of “K” University (the school with which the investigator was affiliated) (approval number 40525-201806-HR-54-01). The principal investigator visited students’ lecture rooms to explain the aims and content of the study, that participation would be voluntary, and that data would be anonymous, and to inform them that they could withdraw from the study at any point without incurring any penalties. All participants provided written consent. The research assistant confirmed participants’ mean score exceeding the median (2.5) for trait anger using the Korean version of the State-Trait Anger Expression Inventory (STAXI).

The effects of the ASCP were measured using a self-report questionnaire featuring a series of measures designed to determine the participants’ levels of trait anger, anger expression style, grateful disposition, and depression. At the beginning of the first session of the program, a pre-intervention test was conducted for both groups using this questionnaire by a trained research assistant. After the pre-test, the intervention group underwent the six-session ASCP and then, upon completion of the program, completed the first post-test; eight weeks after completion of the program, they then performed a second post-test. The control group underwent post-tests at the same time points as the intervention group. For ethical consideration of the control group, a handout summarizing ASCP was provided for the control group after the second post-test.

### 2.6. Data Analysis

Data analysis was performed by IBM SPSS Statistics 25.0 (IBM Corp., Armonk, NY, USA). The general characteristics of the participants were analyzed using frequency and percentage. Authors tested the normality of the data at pre-test and post-test using both the Kolmogorov–Smirnov tests for each group and found that the normality was secured. The homogeneity of the general characteristics and dependent variables in both groups was performed using an independent t test or Chi-square with Fisher’s exact test. A Chi-square with Fisher’s exact test was used when the data were unequally distributed among the cells of the table or when the expected values in any of the cells of a contingency table were below 5. Meanwhile, a repeated-measures analysis of variance (ANOVA) was performed to analyze the differences in the scores of the intervention and control groups in regard to the pre-test, post-test, and follow-up. Statistical significance was established at *p* < 0.05.

## 3. Results

### 3.1. Homogeneity Test Results for General Characteristics and Dependent Variables between Control and Intervention Groups

Female students accounted for 78.6% of the intervention group and 73.3% of the control group, meaning there was no significant difference between the two groups in this regard. Further, there were no significant differences between the two groups in terms of religion, satisfaction with nursing, and perceived interpersonal relationships. In addition, the two groups also showed no significant differences in terms of their trait anger, anger expression style, grateful disposition, or depression scores before the intervention (Table 2 and Table 3).

### 3.2. Comparison of the Changes in Trait Anger, Anger Expression Style, Grateful Disposition, and Depression

A repeated-measures ANOVA was used to compare the trait anger, anger expression style, grateful disposition, and depression of the intervention and control groups at the pre-test, post-test (upon completion of the program), and follow-up (eight weeks after completion of the program).

It was consequently found that trait anger significantly differed between the two groups (F = 12.00, *p* = 0.002) and across time points (F = 56.23, *p* < 0.001). There was a significant interaction between group and time (F = 5.15, *p* = 0.017).

In the analysis of the changes in the three forms of anger expression style, anger-in did not significantly differ between the two groups (F = 0.91, *p* = 0.350) and across time points (F = 3.37, *p* = 0.070). There was a significant interaction between group and time (F = 5.25, *p* = 0.024). Meanwhile, anger-out did not significantly differ between the two groups (F = 0.28, *p* = 0.599), but did significantly differ across time points (F = 22.60, *p* < 0.001), with an interaction existing between group and time (F = 4.62, *p* = 0.014). Finally, anger control significantly differed between the two groups (F = 7.08, *p* = 0.013) and across time points (F = 23.32, *p* < 0.001), with an interaction existing between group and time (F = 7.42, *p* = 0.010).

Grateful disposition significantly differed between the two groups (F = 11.27, *p* = 0.002) and across time points (F = 34.62, *p* = < 0.001), with an interaction existing between group and time (F = 9.33, *p* = 0.003).

With regard to depression, there were no significant differences between the groups and among time points, and there was also no interaction between group and time (Table 4).

## 4. Discussion

The present study developed an ASCP for nursing students and assessed the effects of the program on students’ trait anger, anger expression style, grateful disposition, and depression. We found that, compared to the control group, the intervention group showed a significant reduction in trait anger after the intervention. In a previous study, an anger control program focused on emotion recognition and expression was determined to significantly reduce the level of anger among adolescents [30]. Further, Bilge and Keskin [1] also reported that psychoeducation based on cognitive behavioral techniques significantly reduces anger levels among college students. People who experience their emotions in greater detail can address them more effectively because they have an accurate understanding of them. In particular, negative emotions can be controlled only when they are accurately identified, not through avoiding them. In our study, the intervention group underwent a program that trained them in emotion differentiation through the application of various activities, and they consequently became familiar with various emotions. As a result, they were able to express the particular emotions they felt about others in more detail and to appropriately address them, which presumably reduced their anger levels. A study on Korean college students (using in-depth interviews) [31] reported that students experience anger when they develop unrealistic expectations, feelings of depression, a sense of alienation, disappointment, and frustration. In this context, students who participated in the ASCP are speculated to have shown a reduced level of anger because they identified their distorted thoughts that induced negative emotions and re-evaluated incidents that induced anger.

After participating in the ASCP, the intervention group showed a significant reduction in anger-in and anger-out, the forms of maladaptive anger expression, and a significant increase in anger control, and this suggests that the ASCP was effective for helping the students adopt adaptive means of expressing anger. These results accord with a previous finding that an anger-control program based on cognitive behavioral therapy effectively improves adaptive anger expression [12,15,32]. Our results may be attributable to the fact that participants were able to learn about adaptive anger expression and explore and practice more effective ways of expressing their anger. In addition, empathy training enables individuals to objectively evaluate the validity of others’ as well as one’s own emotions, thoughts, and behaviors; thus, the participants of this study were able to improve their anger expression possibly because they were trained to consider others involved in anger-provoking conflict situations and, concurrently, to objectively evaluate the true nature of their own emotions and thoughts.

In the present study, compared to the control group, the intervention group showed a significant increase in grateful disposition after the program. This is similar to a previous finding in which a gratitude program applying a cognitive behavioral counseling technique was found to effectively increase grateful disposition among college students [33]. We only applied one session of gratitude expression training. Despite the fact that there was only one session, students showed a greater willingness to recognize and express things for which they were grateful because they learned in the previous sessions about how changing thoughts affects negative emotions or behaviors.

Depression did not significantly differ between the two groups in our study. Previous studies have stressed that, in order to significantly improve depression, program contents should directly and specifically address depression-related issues [34]. Further, Kwok, Gu, and Kit [18] reported that positive psychological therapy effectively improves depression symptoms in children with depressive tendencies. Therefore, the results of this study could be attributable to the fact that the ASCP focused on anger, as opposed to depression, and that positive psychology, which has been found to be effective for improving depression, was less emphasized in our program compared to previous programs. On the other hand, considering that depression and anger are significantly associated, modifying the program to address depression more directly may ultimately improve anger management. Lee, Lee, Ahn, Yoo, and Kwon [15] found that an anger-management program integrating cognitive social training and art therapy had a significant impact on both anger and depression; hence, incorporating media that enables more active expression of emotions, such as art activities, may be useful when developing anger-management programs in the future. As all of the participants of the present study were nursing students from the same university, its findings should be generalized with caution. Further, intervention diffusion could not be completely controlled for, as the sample consisted of students of the same grade level in the same school. Hence, future studies should consider using samples from different schools for the intervention and control groups. This study had a small sample size, so the results cannot be generalized; consequently, we recommend further studies be designed with larger sample sizes.

## 5. Conclusions

The ASCP developed in this study could be used as an extracurricular program for nursing students, through which they can alleviate trait anger and increase adaptive anger expression and grateful disposition. This study is meaningful because it presented an effective and specific anger intervention by confirming the applicability of positive psychological traits for anger management. Further, through effective anger management, this study also developed a character education program that facilitates students’ growth of character. However, the ASCP did not have significant effects on the participants’ depression. Therefore, it is necessary to modify the program contents to decrease depression among nursing students.

Based on these findings, we make the following suggestions: Subsequent studies should modify our program and investigate its effects on depression, which were not evident in the present study. Further, the effects of the program on students taking more diverse majors should be examined by developing a systematic manual for the program.

## Figures and Tables

**Table 1 ijerph-18-03059-t001:** Program contents of the anger self-control program (ASCP).

Phase	Themes	Contents
**Understanding anger**	Emotion differentiation	Introduce the program and set group rules Education: What is emotion differentiation? Express emotions with your body in order to become familiar with various emotions Experience the different emotions people feel about the same person Complete the activity report and share your thoughts with everyone Homework: Complete an emotions report every day (in the morning, afternoon, and evening) for a week (describe your emotions in detail).
	Recognizing anger	Education: Understanding anger Express through a drawing how you feel when you are angry Understand the various forms of anger you experience in interpersonal relationships: Same emotions, different situations activity Explore primary emotions related to anger, and describe the reasons such emotion and desires imbued in that emotion Homework: Write about your most recent experience of anger
	Cognitive reconstruction	Education: Cognitive distortion related to anger Identify the positive and negative aspects of anger Identify types of cognitive distortion related to anger Brainstorm thoughts that can mitigate anger Homework: Choose and read one book that helps you in regard to anger management
	Expressing anger in a healthy way	Education: Understanding adaptive anger expression Discover various patterns of anger expression through brainstorming Explore the positive/negative outcomes of your own anger expression pattern Identify effective anger expression methods Apply the “I-message” communication training and practice on your partner Homework: Practice expressing anger in an appropriate manner
**Placating my angry self**	Strengthening positivity	Education: What is positivity? Discovering your strengths that have helped you in difficult situations Gratitude training: Express gratitude to the group members and yourself and share changes that resulted from showing gratitude. Homework: Create a picture book of your strengths
	Controlling my mind to empathize with others	Education: Empathy Empathetic response training Forgive people (including yourself) who have made you angry: Write a letter of forgiveness to express your determination to forgive Evaluation: Share your thoughts on participating in the program

**Table 2 ijerph-18-03059-t002:** Homogeneity test results for general characteristics between control and intervention groups.

Characteristics	Categories	Cont. (*n* = 15)	Int. (*n* = 14)	*x* ^2^	*p*
		*n* (%)			
Gender	Male	4 (26.7)	3 (21.4)	0.11	1.000^†^
	Female	11 (73.3)	11 (78.6)		
Religion	Yes	6 (40.0)	7 (50.0)	0.29	0.715
	No	9 (60.0)	7 (50.0)		
Satisfaction with nursing	Satisfied	8 (53.3)	7 (50.5)	0.47	1.000 ^†^
	Average	4 (26.7)	5 (35.7)		
	Dissatisfied	3 (20.0)	2 (14.4)		
Perceived interpersonal relationships	Good	7 (46.7)	5 (35.7)	0.54	0.877 ^†^
	Average	6 (40.0)	7 (50.0)		
	Poor	2 (13.3)	2 (14.3)		

Cont. = Control group; Int. = intervention group, ^†^ Fisher’s exact test.

**Table 3 ijerph-18-03059-t003:** Homogeneity test results for dependent variables between control and intervention groups.

Variable	Cont. (*n* = 15)	Int. (*n* = 14)	*t*	*p*
	Mean (SD)	Mean (SD)		
Trait anger	3.11 (0.50)	2.91 (0.45)	−1.089	0.286
Anger expression style				
Anger-in	2.20 (0.51)	2.44 (0.38)	1.426	0.165
Anger-out	2.19 (0.44)	2.45 (0.48)	1.534	0.137
Anger-control	2.15 (0.50)	2.01 (0.42)	−0.781	0.442
Grateful disposition	4.28 (0.99)	4.13 (0.67)	−0.444	0.661
Depression	11.00 (8.11)	14.71 (9.43)	1.140	0.264

Cont. = Control group; Int. = intervention group; SD = standard deviation.

**Table 4 ijerph-18-03059-t004:** Comparison of changes in trait anger, anger expression style, grateful disposition, and depression levels between control and intervention groups.

Variables		Pre-Test	Post-Test	Follow-Up	Source	F	*p*
		Mean (SD)	Mean (SD)	Mean (SD)			
Trait anger	Cont.	3.11 (0.50)	2.49 (0.49)	2.55 (0.45)	Group	12.00	0.002
	Int.	2.91 (0.45)	2.01 (0.50)	1.79 (0.35)	Time	56.23	<0.001
					Group × Time	5.15	0.017
Anger expression style							
Anger-in	Cont.	2.20 (0.52)	2.30 (0.57)	2.25 (0.50)	Group	0.91	0.350
	Int.	2.44 (0.38)	2.04 (0.53)	1.85 (0.53)	Time	3.37	0.070
					Group × Time	5.25	0.024
Anger-out	Cont.	2.19 (0.44)	1.93 (0.49)	1.86 (0.44)	Group	0.28	0.599
	Int.	2.45 (0.48)	1.71 (0.38)	1.62 (0.39)	Time	22.60	<0.001
					Group × Time	4.62	0.014
Anger-control	Cont.	2.15 (0.50)	2.28 (0.46)	2.50 (0.43)	Group	7.08	0.013
	Int.	2.01 (0.42)	2.84 (0.53)	3.08 (0.51)	Time	23.32	<0.001
					Group × Time	7.42	0.010
Grateful disposition	Cont.	4.28 (0.99)	4.62 (0.78)	4.95 (0.76)	Group	11.27	0.002
	Int.	4.13 (0.67)	5.67 (0.42)	6.00 (0.45)	Time	34.62	<0.001
					Group × Time	9.33	0.003
Depression	Cont.	*11.00 (8.11)*	9.67 (7.21)	8.73 (6.11)	Group	0.11	0.747
	Int.	*14.71 (9.43)*	8.93 (5.34)	7.86 (4.91)	Time	5.97	0.019
					Group × Time	1.77	0.194

## Data Availability

The data used and/or analyzed during the current study are available from the corresponding author on request.

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
