# Peer review of "Effects of an Extracurricular Anger Self-Control Program for Nursing Students"

_ijerph, 2021, doi:10.3390/ijerph18063059_

Round 1

Reviewer 1 Report

First of all, I would like to congratulate the authors for the work presented. for the work presented, a quasi-experiment is a very valid design in the educational context. valid design in the educational context.

Reviewing the theoretical framework, it would be good if the authors could expand the references, as there is an extreme poverty in the theoretical foundation of the article, which detracts from its validity. It should be noted that from 2019 to 2021 there is no reference, from 2018 there are two references, and from 2016 and 2017 three references from each year, it is important to review this.

In relation to the research design, I have to say that it is correct, the dependent and independent variables are established. It could be expanded how they were controlled, or at least attempted to control, extraneous variables.

The data analysis was well done, however, it is not detailed if there is normality in the distribution of the data, an ANOVA is used, and a chi-square, but a Student's t-test should have been used, to determine if the difference obtained was normal.
Student's t test should have been used to determine whether the difference obtained in the results is due to chance or to the treatment administered.

Otherwise, it is a very interesting article, and once again I congratulate the authors.

Author Response

Thank you for the review.

Reviewer 2 Report

In this work, Jun and colleagues explain and validate a new program for anger self-control. This program or something similar, in my opinion, is very important and it should be used in all universities and not only for nursing students. I agree with the authors: this program may be very important in Korean society for the reasons they provide. Anger management is important in all professions and mostly in healthcare professions. I think that the ASCP is well explained and well done. 

My only recommendations are about the number of subjects involved in this study: maybe in a future work authors could involve a bigger amount of participants in the boh control group and study group. 
I suggest changing Table 1's layout because it is hard to read in the "themes" section.

In the end, I think that this study is very interesting and has high scientific soundness and it is well written.

Author Response

Thank you for your review.
